# Coenzyme Q10 Supplementation Increases Removal of the *ATXN3* Polyglutamine Repeat, Reducing Cerebellar Degeneration and Improving Motor Dysfunction in Murine Spinocerebellar Ataxia Type 3

**DOI:** 10.3390/nu14173593

**Published:** 2022-08-31

**Authors:** Yu-Ling Wu, Jui-Chih Chang, Hai-Lun Sun, Wen-Ling Cheng, Yu-Pei Yen, Yong-Shiou Lin, Yi-Chun Chao, Ko-Hung Liu, Ching-Shan Huang, Kai-Li Liu, Chin-San Liu

**Affiliations:** 1Vascular and Genomic Center, Institute of ATP, Changhua Christian Hospital, Changhua 50091, Taiwan; 2Cardiovascular and Mitochondrial Related Disease Research Center, Hualien Tzu Chi Hospital, Buddhist Tzu Chi Medical Foundation, Hualien 97004, Taiwan; 3Center of Regenerative Medicine and Tissue Repair, Changhua Christian Hospital, Changhua 50091, Taiwan; 4General Research Laboratory of Research Department, Changhua Christian Hospital, Changhua 50091, Taiwan; 5School of Medicine, Chung Shan Medical University, Taichung 40203, Taiwan; 6Department of Pediatrics, Division of Allergy, Asthma and Rheumatology, Chung Shan Medical University Hospital, Taichung 40203, Taiwan; 7Department of Nutrition, Chung Shan Medical University, Taichung 40203, Taiwan; 8Inflammation Research & Drug Development Center, Changhua Christian Hospital, Changhua 50091, Taiwan; 9Department of Nutrition, Chung Shan Medical University Hospital, Taichung 40203, Taiwan; 10Department of Neurology, Changhua Christian Hospital, Changhua 50094, Taiwan; 11Graduate Institute of Integrated Medicine College of Chinese Medicine, China Medical University, Taichung 40447, Taiwan; 12College of Medicine, National Chung Hsing University, Taichung 40227, Taiwan

**Keywords:** coenzyme Q10, spinocerebellar ataxia type 3, locomotor functions, Purkinje cells, muscle atrophy

## Abstract

Coenzyme Q10 (CoQ10), a well-known antioxidant, has been explored as a treatment in several neurodegenerative diseases, but its utility in spinocerebellar ataxia type 3 (SCA3) has not been explored. Herein, the protective effect of CoQ10 was examined using a transgenic mouse model of SCA3 onset. These results demonstrated that a diet supplemented with CoQ10 significantly improved murine locomotion, revealed by rotarod and open-field tests, compared with untreated controls. Additionally, a histological analysis showed the stratification of cerebellar layers indistinguishable from that of wild-type littermates. The increased survival of Purkinje cells was reflected by the reduced abundance of TUNEL-positive nuclei and apoptosis markers of activated p53, as well as lower levels of cleaved caspase 3 and cleaved poly-ADP-ribose polymerase. CoQ10 effects were related to the facilitation of the autophagy-mediated clearance of mutant ataxin-3 protein, as evidenced by the increased expression of heat shock protein 27 and autophagic markers p62, Beclin-1 and LC3II. The expression of antioxidant enzymes heme oxygenase 1 (HO-1), glutathione peroxidase 1 (GPx1) and superoxide dismutase 1 (SOD1) and 2 (SOD2), but not of glutathione peroxidase 2 (GPx2), were restored in 84Q SCA3 mice treated with CoQ10 to levels even higher than those measured in wild-type control mice. Furthermore, CoQ10 treatment also prevented skeletal muscle weight loss and muscle atrophy in diseased mice, revealed by significantly increased muscle fiber area and upregulated muscle protein synthesis pathways. In summary, our results demonstrated biochemical and pharmacological bases for the possible use of CoQ10 in SCA3 therapy.

## 1. Introduction

Spinocerebellar ataxia type 3 (SCA3)/Machado–Joseph disease, one of the harmful, late-onset polyglutamine (PolyQ) diseases, is the most prevalent autosomal dominant spinocerebellar ataxia worldwide [1]. Abnormal cytosine–adenine–guanine (CAG) repeats in the *ATXN3* gene coding region, resulting in a polyQ expansion containing 60–87 repeats in the mutant ataxin-3 protein, enhances mitochondrial apoptosis, leading to neuronal death in specific brain regions of SCA3 patients [1,2]. Clinical neurodegenerative syndromes of SCA3 patients include progressive gait imbalance and loss of coordination, muscle atrophy, dystonia, ophthalmoplegia and progressive cerebellar ataxia [3,4]. Although no promising treatment has been developed for SCA3, a potential therapeutic strategy to retard disease progression includes the down-regulation of the pathological mechanism by supporting cerebellar neuron function and reducing aggregates or accumulations of mutant protein and oxidative stress [5,6].

Diminished capacity to deal with the oxidative stress induced by reactive oxygen species (ROS) leads to neuronal apoptosis in several neurodegenerative disorders, including PolyQ diseases [7,8]. In the leukocytes of SCA3 patients, the accumulation of nuclear and mitochondrial DNA damage was associated with a strong increase in oxidative stress [9]. Moreover, enhanced sensitivity to oxidative challenges and impaired antioxidant defenses caused by the decreased expression of antioxidant enzymes were found in the brain and leukocytes of SCA3 patients, as well as in cell and *Drosophila* models of SCA3 [9,10,11]. Supplemented antioxidant capacity has been a common therapeutic SCA3 treatment, employing various antioxidant agents or herbal medicines such as curcumin, caffeic acid, or resveratrol in different SCA3 models in vitro and in vivo [11,12,13].

Autophagy is the major self-degradation system and contributes significantly to cellular energy metabolism [14]. It is responsible for intracellular protein clearance and regulating beclin-1, p62 and microtubule protein 1 light chain 3 (LC3) to control autophagosome formation and degradation. Facilitating autophagy influx through increased autophagosome biogenesis or autophagosome fusion with lysosomes promotes protein clearance [14]. Autophagy impairment has been found in human SCA3 patients [15], and has been consistently validated in cell and transgenic mouse models of SCA3 [16,17]. Autophagy induced with pharmaceuticals or the over-expression of autophagy protein beclin-1 reduces the neurotoxic accumulation of mutant ataxin-3 protein and is recognized as an SCA3 treatment [18,19]. The foregoing studies confirm a crucial role of autophagy in restraining SCA3 disease progression and suggest its therapeutic potential for other PolyQ diseases.

Coenzyme Q10 (CoQ10) is an essential component of the mitochondrial electron transport chain to generate adenosine triphosphate (ATP) in cells. Additionally, CoQ10, the only endogenously synthesized lipid-soluble antioxidant, prevents free-radical-induced oxidative damage to proteins, lipoproteins, mitochondrial DNA and cellular membranes [20]. Notably, beyond its antioxidant/radical-scavenging activity, CoQ10 may influence gene expression related to cell signaling, metabolism, nutrient transport [21] and inflammation [22]. Because of its pleiotropic functions with good safety and tolerance, CoQ10 may be useful as a nutritional supplement for a variety of pathological conditions, including neurodegenerative diseases [23]. The efficacy of CoQ10 treatment for Huntington’s disease (HD), Parkinson’s disease (PD) and Amyotrophic lateral sclerosis (ALS) is still controversial, especially for SCA, which few studies have addressed [24].

Supplementation with CoQ10 improved clinical profiles in SCA1 and SCA3 patients [25]. A dose-dependent effect of daily CoQ10 on decreasing scale for the assessment and rating of ataxia (SARA) scores was demonstrated in SCA3 patients [25], suggesting the importance of clarifying regulatory mechanisms of SCA3 pathology susceptible to CoQ10 treatment. Moreover, although a reduction of mutant ataxin-3 aggregations was evident in CoQ10-treated SCA3 cells [26], there is still no data confirming its effect on SCA3. Therefore, considering its potential therapeutic value, we assessed the protective effects of CoQ10 and underlying mechanisms in delaying SCA3 pathology with high-dose CoQ10 supplementation in early onset of SCA3 in transgenic mice.

## 2. Materials and Methods

### 2.1. Materials

CoQ10 was provided by Metagenics, Inc. (Aliso Viejo, CA, USA). Antibodies against heat shock protein 27 (Hsp27), p53, Beclin-1 and glyceraldehyde 3-phosphate dehydrogenase (GAPDH) were from Santa Cruz Biotechnology (Santa Cruz, CA, USA). Specific primary antibodies for total (pro) and cleaved poly ADP-ribose polymerase (PARP), and phosphorylated p53 and serine/threonine kinase (AKT) were from Cell Signaling Technology Inc. (Beverly, MA, USA). Antibodies against caspase-3, LC3, heme oxygenase 1 (HO-1) and myosin heavy chain (MyHC) were from Novus Biologicals (Littleton, CO, USA), MBL (San Diego, CA, USA), BD Biosciences (Boston, MA, USA) and eBioscience (San Diego, CA, USA), respectively. Antibodies against p62 and ataxin-3 were obtained from Abcam (Cambridge, UK). Specific antibodies for superoxide dismutase (SOD)1, SOD2, glutathione peroxidase (GPx) 1 and GPx2 were obtained from Genetex Inc. (GeneTex, CA, USA).

### 2.2. Animal Model

All animals were purchased from the Jackson Laboratory (Bar Harbor, ME, USA). SCA3 transgenic mice harbored a yeast artificial chromosome transgene expressing the human *ATXN3* gene with a pathological polyglutamine tract expanded 84-CAG-repeat motif (84Q). Those mice were generated and maintained on the same C57BL/6J background strain. Genotyping of 84Q SCA3 transgenic mice was confirmed by PCR with DNA isolated from a mouse tail biopsy at the time of weaning [27]. Mice were housed under a 12 h light–dark cycle with free access to food and water. All animal experiments were performed with approval of the Institutional Animal Care and Use Committee of Chung Shan Medical University for the care and use of laboratory animals. A total of 21 experimental mice were randomly divided into three groups: wild-type (WT) mice (*n* = 7), 84Q SCA3 mice with chow diet (*n* = 7) and 84Q SCA3 + CoQ10 (*n* = 7). Mice at the postnatal age of 9 months (early onset stage) [28] were fed a regular chow diet mixed with vehicle or CoQ10 (1000 mg/kg/day) for 7 months and were then sacrificed with CO_2_ asphyxiation. The cerebellum and muscle tissues were removed and frozen immediately.

### 2.3. Plasma CoQ10 Measurement

Blood was collected when mice were sacrificed at 16 months of age. For use, blood samples were allowed to thaw at 4 °C and centrifuged. Plasma levels of CoQ10 were determined according to methods described previously [29,30]. Samples of plasma were deproteinated with methanol and n-hexane. This mixture was vortexed 5 min and then centrifuged at 2500× *g* for 20 min at 4 °C. Subsequently, the clear n-hexane layer was collected into another microcentrifuge tube, and the n-hexane extraction was repeated once more. Plasma extracts were evaporated to dryness using a stream of nitrogen gas. The dry residue was dissolved in mobile phase, filtered through a PVDF filter (4 mm, 0.45 μm; Millipore, Bedford, MA, and injected into the high-performance liquid chromatography (HPLC) system. A microBondapak C18 3.9 mm × 30-cm stainless steel column with a guard column 3 × 22 mm packed with microBondapak C18 was used with a mobile phase of methanol-n-hexane 85:15 (vol/vol). The flow rate was 1 mL/min, and the detector wave-length was 276 nm. The amounts of CoQ10 in plasma were identified according to a calibration curve that was constructed by plotting peak area vs. concentration. Linearity was achieved in the concentration range of 0.12 to 1.92 μg/mL.

### 2.4. Rotarod Test

Motor coordination and balance were measured using a rotarod apparatus (Accelerating Model, Ugo Basile Biological Research Apparatus, Varese, Italy) [31]. Mice were placed on the rotarod at a constant speed of 5 rpm, which was then accelerated from 5 to 40 rpm for 300 s. Mean latency was defined as the number of seconds that a mouse remained on the rod. Mice were permitted to rest for at least 15 min before the next test to avoid fatigue. Each mouse underwent five rotarod tests, and the latency was recorded and analyzed statistically. The rotarod test was performed after 7 months of treatment.

### 2.5. Open Field Behavior Assay

To evaluate exploratory and general motor behavior, mice were subjected to an open field test after 7 months of CoQ10 supplementation. Mice were placed individually in a 30 cm × 18.8 cm × 13.5 cm home cage-like apparatus and allowed to freely explore the arena for 10 min. Exploratory behavior was recorded with an overhead tracking system Top Scan (CleverSys, Reston, VA, USA) equipped with an active infrared-sensitive camera. Top Scan Light (CleverSys, Reston, VA, USA) tracking software was used to calculate the distance, velocity, zone change frequency and total number of entries into the center area of the open field [32]. The testing apparatus was exhaustively cleaned with 70% ethanol between tests to reduce possible bias due to odors left by other mice.

### 2.6. Preparation of Histological Tissue Sections

At 16 months of age, mouse cerebella were harvested and fixed with 4% paraformaldehyde for 48 h, followed by dehydration. Tissues were depleted of fats by soaking them in xylene and infiltrating them with melted paraffin in a mold. After the paraffin hardened, embedded tissue was removed from the mold. Tissues were sectioned to 3 μm using a sledge microtome and transferred to slides.

### 2.7. Histological Staining

For hematoxylin and eosin staining, cerebellar specimens were de-paraffinized and rehydrated sequentially by soaking them in xylene and then in 100, 95, 80 and 75% alcohol and water. Afterward, they were stained with hematoxylin and eosin (H&E) following standard procedures [28]. Slides were photographed using an Olympus optical microscope (Olympus, Tokyo, Japan). For each sagittal section, six fields covering the entire cerebellar cortex were digitized, and cerebellar molecular layers (ML), granular layers (GL) and Purkinje cells were analyzed using Image J software (National Institutes of Health, NIH). The number of surviving Purkinje cells was counted in the lobules between the ML and the GL of cerebellar slices. Muscle specimens from hind legs were fixed in 10% formalin for 24 h at room temperature. Then they were stained with H&E following standard procedures [33]. H&E-stained muscle tissue was photographed at 200X magnification. The muscle fibers’ cross-sectional area (CSA) was calculated from 100 tibialis (TA) muscle fibers in each of 5 random fields using an AlphaImager 2200 (Alpha Innotech Corporation, San Leandro, CA, USA). The CSA was determined as the mean of 100 TA fibers. Then the CSAs were categorized: <1500, 1500–2000, 2000–2500, >2500 μm^2^. Data were expressed as the percentage of 100 TA muscle fibers per group.

The immunohistochemical staining of brain sections was performed with an anti-ataxin-3 antibody (1:300 dilution; Millipore, Billerica, MA, USA) as described [34]. Sections stained by the horseradish peroxidase (HRP)-conjugated secondary antibody were detected by using 3,3′-diaminobenzidine (DAB; Sigma) and then visualized with a microscope at 40× magnification. The expression of ataxin-3 in Purkinje cells was analyzed using Image J software, and 50 cells were counted for quantification purposes.

### 2.8. Terminal Deoxynucleotidyl Transferase-Mediated dUTP Nick-End Labeling (TUNEL) Staining

Cerebella were fixed in 4% paraformaldehyde overnight, and then they were paraffin-embedded and sectioned with a sledge microtome. Cerebellar sagittal sections (3 μm) were examined to detect apoptosis using a TUNEL detection kit (Roche, Meylan, France) following the manufacturer’s instructions. Sections were examined under the microscope, and six randomly selected fields of TUNEL-positive apoptotic cells were analyzed using Image J software [35]. The percentage of TUNEL-positive signals was calculated in 100 cells of each mouse.

### 2.9. Protein Extraction and Western Blot

Phosphate-buffered saline and RIPA lysis buffer were used to prepare total protein extracts from homogenized cerebella. Homogenates were centrifuged at 14,000× *g* for 30 min at 4 °C, and supernatant protein concentrations were quantified using a modified Lowry assay [36]. Equal amounts of protein were denatured and separated by SDS-PAGE and then transferred onto polyvinylidene difluoride membranes (New Life Science Product, Inc., Boston, MA, USA). Membranes were blotted sequentially with primary antibodies and horseradish peroxidase-conjugated secondary antibodies (Bio-Rad, Hercules, CA, USA), which were developed using an enhanced chemiluminescence kit (Perkin–Elmer Life Science, Boston, MA, USA). Immunoblots were visualized using a luminescent image analyzer (LAS-1000 plus, Fuji Photo Film Company, Japan) and quantified with AlphaImager 2200 (Alpha Innotech Corp., San Leandro, CA, USA).

### 2.10. Statistical Analysis

Experimental data were presented as means ± SD and calculated from a mix of male and females of mice. Differences among experimental groups were evaluated using one-way ANOVA and two-tailed Student’s *t* tests. *p* < 0.05 was considered statistically significant.

## 3. Results

### 3.1. Effects of CoQ10 on Plasma Concentration in 84Q SCA3 Mice

After 7 months of CoQ10 treatment (1000 mg/kg/day), plasma CoQ10 was 121.4 nM in 84Q SCA3 mice 16 months of age (Figure 1). There are no detection plasma CoQ10 levels in 84Q SCA3, or in wild-type control mice. These results show that oral CoQ10 achieves effective plasma bioavailability in 84Q SCA3 mice.

### 3.2. Effects of CoQ10 on Rotarod Performance and Open-Field Behavior in 84Q SCA3 Mice

Figure 2A shows the accelerated rotarod balance test that was conducted to assess the motor coordination of wild-type controls and 84Q SCA3 mice. After 7 months of supplementation with CoQ10, mice remained on the rotarod longer than 84Q SCA3 mice (Figure 2A; *p* < 0.05). Treated mice showed continual improvement of motor function throughout the 7-month period. 84Q SCA3 mice showed a dramatic decrease in locomotor activity during the 10-min open-field behavior assay (Figure 2B). Compared with wild-type control mice, 84Q SCA3 mice travel less in a 10-min period (Figure 2C; *p* < 0.05). A significant improvement in travel distance, average velocity (Figure 2D; *p* < 0.05), frequency of zone change (Figure 2E; *p* < 0.05) and number of entries into the central area (Figure 2F; *p* < 0.05) was observed in 84Q SCA3 mice supplemented with CoQ10. These results show that oral CoQ10 retards 84Q SCA3 disease progression and improves motor function in 84Q SCA3 mice.

### 3.3. Effects of CoQ10 on the Granular and Molecular Layers of the Cerebellar Cortex in 84Q SCA3 Mice

After 7 months of CoQ10 treatment, the average thickness of the GL in 84Q SCA3 mice was significantly reduced compared to that of wild-type control mice but was not significantly rescued (Figure 3A,B; *p* < 0.05). 84Q SCA3 mice revealed a significant reduction in the average thickness of the ML compared with wild-type control mice. The addition of CoQ10 helped to preserve ML thickness in 84Q SCA3 mice (Figure 3C,D; *p* < 0.05).

### 3.4. Effects of CoQ10 on Cerebellar Apoptosis in 84Q SCA3 Mice

Compared with wild-type control mice, 84Q SCA3 mice not only showed a loss of Purkinje cells but also increases in phosphorylated p53 and the frequency of apoptosis in cerebellar Purkinje cells, as measured with H&E staining and TUNEL analysis, respectively (Figure 4A–C; *p* < 0.05). The administration of CoQ10 restored the number of Purkinje cells to normal levels and reduced the expression of phosphorylated p53 and TUNEL signal in the cerebellar Purkinje cells of 84Q SCA3 mice (Figure 4A–C; *p* < 0.05). Interesting, CoQ10 significantly increased Purkinje cells number in comparison to wild-type control mice. In addition, compared with wild-type control mice, cleaved caspase 3 and PARP expression were augmented in the cerebellum of 84Q SCA3 mice, and this increase in expression was significantly diminished by CoQ10 supplementation (Figure 4A; *p* < 0.05).

### 3.5. Effects of CoQ10 on Mutant Ataxin-3, Hsp27, Autophagy and Antioxidant Protein Expression in 84Q SCA3 Mice

Compared with wild-type control mice, mutant ataxin-3 protein was increased, and Hsp27 protein was reduced in the cerebellum of 84Q SCA3 mice. CoQ10 supplementation diminished exogenous mutant ataxin-3 expression and increased Hsp27 expression in 84Q SCA3 mice (Figure 5A; *p* < 0.05). SCA3 mice indeed had a significantly higher performance of ataxin-3 protein in the Purkinje cells of the cerebellar cortex than those of wild-type control mice (Figure 5B; *p* < 0.05). The quantification also showed a consistent result (Figure 5B; right panel). The significantly reduced expression of ataxin-3 in Purkinje cells was found in the CoQ10 group relative to the non-treated 84Q SCA3 group (Figure 5B; *p* < 0.05), and it was no difference from the control of the wild-type group. Compared with wild-type control mice, autophagy levels in 84Q SCA3 mice as measured by the expression of p62, Beclin 1 and LC3-II in the cerebellum were significantly decreased. The supplementation of CoQ10 augmented autophagy levels in Q84 SCA3 mice (Figure 5C; *p* < 0.05). Compared to wild-type control mice, antioxidant capacity was damagingly low in 84Q SCA3 mice, as evidenced by the reduced expression of HO-1, GPx2, SOD1 and SOD2 in the cerebellum. CoQ10 supplementation markedly increased the expression of HO-1, GPx1, SOD1 and SOD2 in 84Q SCA3 mice (Figure 5D; *p* < 0.05). However, the CoQ10-raised Hsp27, HO-1, and GPx1 protein expression of 84Q SCA3 mice were better than these in wild-type control mice.

### 3.6. Effects of CoQ10 on Skeletal Muscle Atrophy of 84Q SCA3 Mice

Compared with wild-type control mice, 84Q SCA3 mice showed decreased ratios of muscle mass and body weight of quadriceps, gastrocnemius, tibialis, extensor digitorum longus and soleus muscles. When 84Q SCA3 mice supplemented with CoQ10, these were significantly increased (Table 1). There was a significant decrease of TA fiber area (CSA) and size in 84Q SCA3 mice compared to WT mice, with 100% of TA muscle fibers having a CSA < 1500 μm^2^ (Figure 6A,B; *p* < 0.05). CoQ10 treatment was associated with increased sizes of regenerated TA muscle fibers, whose percent distribution is shown in Figure 6C. The expression of MyHC in gastrocnemius (GA) muscle was also significantly increased in CoQ10-treated 84Q SCA3 mice in comparison with 84Q SCA3 mice (Figure 6D; *p* < 0.05), although not to the level of wild-type control mice. Although the expression of phosphorylated AKT in GA muscle was significantly increased in wild-type control mice in comparison with 84Q SCA3 mice, CoQ10 treatment was not statistically significant for 84Q SCA3 mice and the quantification shown as the down panel of Figure 6D.

## 4. Discussion

Loss of motor coordination and alteration of cerebellar architecture caused by the loss of Purkinje cells are common features of SCA3 patients and mice [37,38]. In this study, after 7 months of oral CoQ10 administration, a remarkable reduction of neurological damage was achieved in early SCA3 mice, revealed by increasing locomotor ability and by the reversed degeneration of the cerebellar Purkinje layer, the granule cell layer and the molecular layer. Furthermore, CoQ10 stimulated AKT pathway-mediated protein synthesis to increase MyHC expression [39] in the hindlimb muscles of SCA3 mice, by increasing skeletal muscle mass to improve the homeostatic balance between muscle protein synthesis and degradation [40]. Muscle atrophy is observed mainly in the extremities of SCA3 patients, who experience a decrease of muscle mass in distal regions, with a great reduction of thigh and calf muscle circumference, leading to locomotor disabilities and disrupted gait [41]. SCA3 patients display greater weight loss and muscle mass reduction compared to patients of type 10 group of SCA [42]. Similar hindlimb instability and muscle loss was also observed in the SCA3 mouse model, consistent with other findings [43]. Thus, a reversal of skeletal muscle atrophy could be a goal of SCA3 treatment, to enhance patient quality of life.

Tumor suppressor p53, a transcription factor, regulates DNA repair, cell cycle and apoptosis [44]. The uppregulation of p53 contributes to the intrinsic mitochondrial-mediated apoptosis induced by caspases 9, 3, 7 and PARP. It results in apoptotic neuronal death caused by the degradation of nuclei and cytoskeletal proteins, DNA fragmentation and apoptotic body formation [43]. In SCA3 mice, mutant ataxin-3 protein induces the de-ubiquitylation and stabilization of p53 to trigger the expression of pro-apoptotic molecules, like Bax, in the cerebellum. Moreover, it also facilitates p53 activation to induce the expression of activated caspase-3 and subsequent apoptotic neuronal death in cerebellar and pontine nucleus neurons [44]. This process is blocked by the p53 inhibitor, pifithrin-α [45]. This explains our finding that CoQ10 reduced neuronal death in the SCA3 cerebellum, related to the reduced p53 activation of the intrinsic mitochondria-mediated apoptosis induced by caspase 3 and PARP and clearance of mutant ataxia-3 proteins. As a consequence, CoQ10 increases levels of antioxidant proteins, including HO-1, GPx1 and SOD2, supporting the findings of previous studies, showing the oxidative-damage-induced neuropathological impairment of SCA3 caused by the accumulation of mutant ataxin-3 [9,10,46].

Autophagy is a degradation pathway not only for mutant, aggregate-prone proteins in SCA3 [15] but also for other PolyQ diseases, such as Alzheimer’s disease (AD), HD and PD [47]. Enhancing autophagy through a diversified strategy including physiotherapy [16], pharmacotherapy [19] and herbal medicines [48] is a useful therapeutic intervention for clearing mutant ataxia-3 protein to retard or prevent the progression of SCA3. CoQ10 protein of acute myocardial ischemia–reperfusion injury acts through the enhancement of antioxidative capacity and autophagy pathways [49] but is still unknown in SCA3. In the present study, we showed that the CoQ10 neuroprotection is related to remarkably increased autophagy, revealed by the upregulation of autophagy markers of p62, Beclin-1 and LC3-II proteins in consequence of reduced levels of mutant ataxin-3 protein. This is supported by the restoration of antioxidative capacity and a simultaneous increase of Hsp27 expression. Hsp27, a powerful ATP-independent chaperone, not only inhibits the accumulation of denatured proteins and promotes their refolding but also exhibits strong anti-cell-death activity at multiple points of the apoptotic signaling pathway [5,50]. The reduction of Hsp27 in the early stages of SCA3 helps to trigger disease onset, via an increase of ROS [5]. The overexpression of Hsp27 suppresses cell death induced by the expression of expanded PolyQ via the downregulation of oxidative stress [6]. Thus, our results consistently link the neuroprotection offered by CoQ10 with the elimination of mutant ataxin-3-induced cytotoxicity.

CoQ10 is able to cross the blood–brain barrier and significantly increases cerebral cortex mitochondrial concentrations in rats for up to 12 months [51]. Human subjects suffered no adverse effects such as nausea, at doses of CoQ10 up to 3000 mg/day [52]. Clinical reports have shown that oral CoQ10 is safe and tolerated at doses up to 3000 mg/day in PD patients [53] and in amyotrophic lateral sclerosis patients [54]. Therefore, high-dose CoQ10 therapy in Friedreich’s ataxia patients improved International Co-operative Ataxia Ratings Scale (ICARS) and the cardiac and skeletal muscle bioenergetics throughout the 47-month trial [55]. The effect of high-dose oral CoQ10 has not yet been adequately investigated in SCA3. In transgenic mice with AD, oral supplementation with high-dose CoQ10 at 1000 mg/kg/day not only increased antioxidant capacity [56] but also prolonged survival, improved motor performance and reduced Huntington aggregation [57]. Moreover, CoQ10 serum concentrations at day 30 after administration reached 133 nM, confirming adequate drug absorption [57]. We obtained similar results in 84Q SCA3 mice. After 7 months of the oral administration of 1000 mg CoQ10/kg/day, plasma CoQ10 concentrations were 121.4 nM. Thus, it consistently appears that high-dose CoQ10 may have therapeutic benefits for the treatment of PolyQ diseases, including SCA3. Moreover, based on our previous finding that oral CoQ10 suppresses disease progression in SCA type 1 mice [34], we found that CoQ10 can amplify rotarod performance and motor activity benefits of exercise therapy in 84Q SCA3 mice (Appendix A). Thus, combined oral CoQ10 and exercise in SCA3 therapy seems worthy of further investigation in order to explore its therapeutic efficacy in PolyQ diseases.

## 5. Conclusions

In summary, our results demonstrated that oral CoQ10 has the biochemical and pharmacological effects in SCA3 therapy.

## Figures and Tables

**Figure 1 nutrients-14-03593-f001:**
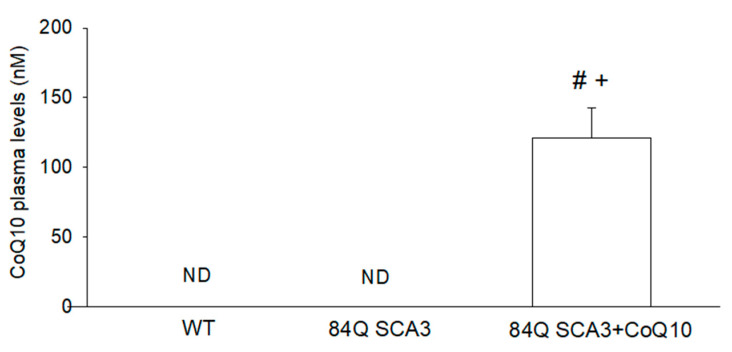
Effects of CoQ10 on plasma concentration in 84Q SCA3 mice. At the age of 16 months, plasma levels of CoQ10 in WT, 84Q SCA3 and CoQ10-treated 84Q SCA3 mice were measured by HPLC. Data are presented as the mean ± SD. # *p* < 0.05 indicates the 84Q SCA3 + CoQ10 group compare to the 84Q SCA3 group. + *p* < 0.05 indicates the 84Q SCA3 + CoQ10 group compared to the WT group. WT, wild-type controls; 84Q SCA3, chow diet of vehicle; 84Q SCA3 + CoQ10, CoQ10 supplementation. ND, not detected.

**Figure 2 nutrients-14-03593-f002:**
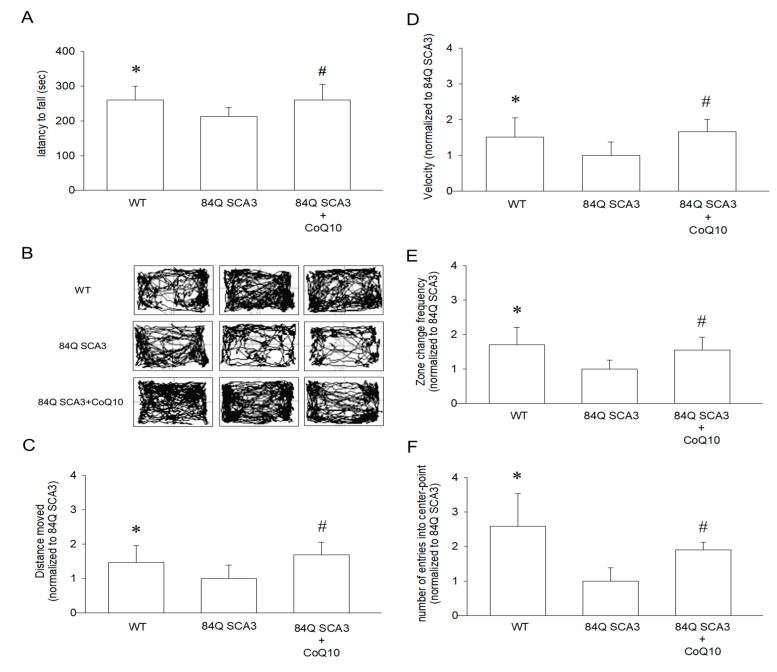
Effects of CoQ10 on rotarod performance and open-field behavior in 84Q SCA3 mice. At the age of 16 months, (**A**) we performed an accelerating rotarod test with all three groups of mice after 7 months of treatment and measured their persistence on the rotarod until a fall. (**B**) All groups of mice were subjected to the open-field behavior assay after the treatment group had received 7 months of treatment. Movements were assessed using the overhead tracking system, Top Scan. Representative traces of spontaneous movement in an open field behavior test during a 10 min monitoring period are shown. Open-field behavior was quantified to determine the (**C**) distance moved, (**D**) the average velocity, (**E**) the zone change frequency and (**F**) the number of entries into the center area. In panels (**C**–**F**), values from treated mice were normalized to those of the 84Q SCA3 group. Data are presented as the mean ± SD. * *p* < 0.05 indicates the WT group compared to the 84Q SCA3 group. # *p* < 0.05 indicates the 84Q SCA3 + CoQ10 group compared to the 84Q SCA3 group. WT, wild-type controls; 84Q SCA3, chow diet of vehicle; 84Q SCA3 + CoQ10, CoQ10 supplementation.

**Figure 3 nutrients-14-03593-f003:**
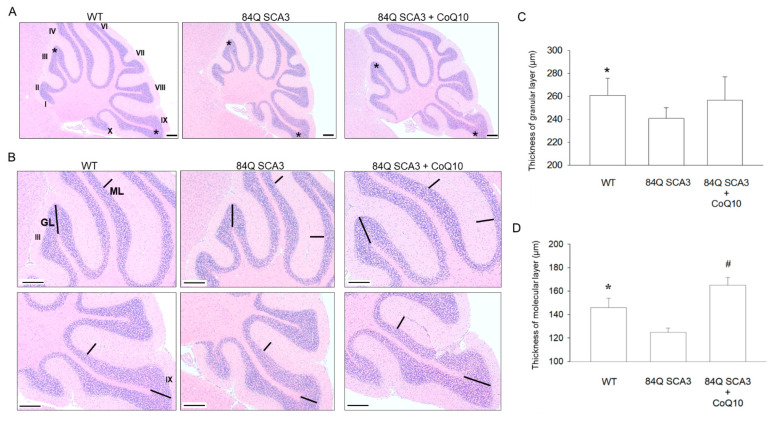
Effects of CoQ10 on the granular (GL) and molecular layers (ML) of the cerebellar cortex in 84Q SCA3 mice. (**A**) Graphs represent the H&E staining of the sagittal section of the cerebellum near the junction of the vermis (approximate 1 mm) at the age of 16 months. Histological sections of the cerebellar vermis show the stereotypical foliation pattern and the asterisk in III and IX lobules indicate the sampling area. (**B**) Magnification images of the sampling area shows that the GL thickness of cerebellar tissue was determined from the distance from the tip of the GL to the white matter, in contrast to the ML thickness, determined by the distance from the GL to the edge of the ML, which is indicated by the black line. (**C**) Quantification of the average thickness of the GL and (**D**) ML. Data are presented as the mean ± SD. Scale bars, 200 μm. * *p* < 0.05 indicates the WT group compared to the 84Q SCA3 group. # *p* < 0.05 indicates the 84Q SCA3 + CoQ10 group compared to the 84Q SCA3 group. WT, wild-type controls; 84Q SCA3, chow diet of vehicle; 84Q SCA3 + CoQ10, CoQ10 supplementation.

**Figure 4 nutrients-14-03593-f004:**
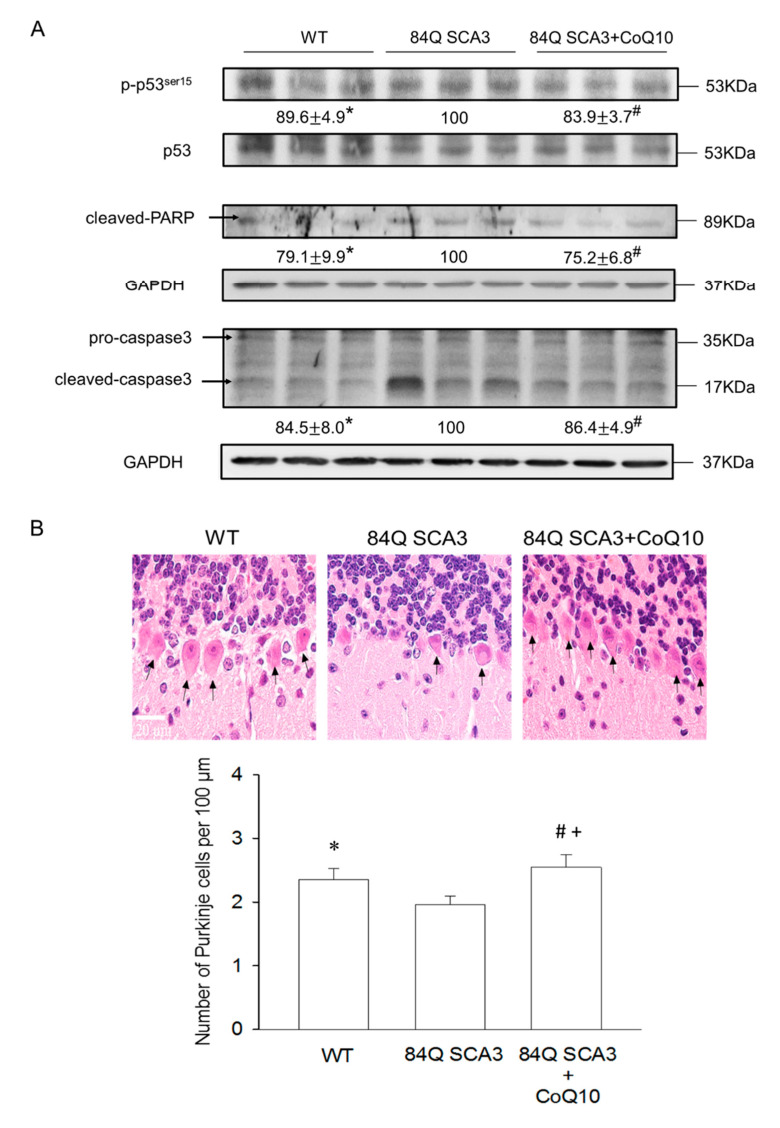
Effects of CoQ10 on cerebellar apoptosis in 84Q SCA3 mice. At the age of 16 months of mice, Western blot analysis of total protein extracts of cerebellar homogenates. (**A**) Protein expression of phosphorylated p53 normalized internal control (total p53) and cleaved caspase 3 and PARP normalized against an internal control (GAPDH), with the same amount of loading protein. (**B**) H&E staining of cerebellar Purkinje cells. The arrows indicating Purkinje cells were identified in the Purkinje cell layer between the molecular layer and granular layer. Average Purkinje cell numbers were calculated per 100 μm. (**C**) Arrows indicate positive signals of brown TUNEL staining in cerebellar Purkinje cells. TUNEL signals were quantified per 100 Purkinje cells. In panels A, values from treated mice were normalized to those of 84Q SCA3 mice. Data are presented as the mean ± SD. * *p* < 0.05 indicates the WT group compared to the 84Q SCA3 group. # *p* < 0.05 indicates the 84Q SCA3 + CoQ10 group compared to the 84Q SCA3 group. + *p* < 0.05 indicates the 84Q SCA3 + CoQ10 group compared to the WT group. WT, wild-type controls; 84Q SCA3, chow diet of vehicle; 84Q SCA3 + CoQ10, CoQ10 supplementation.

**Figure 5 nutrients-14-03593-f005:**
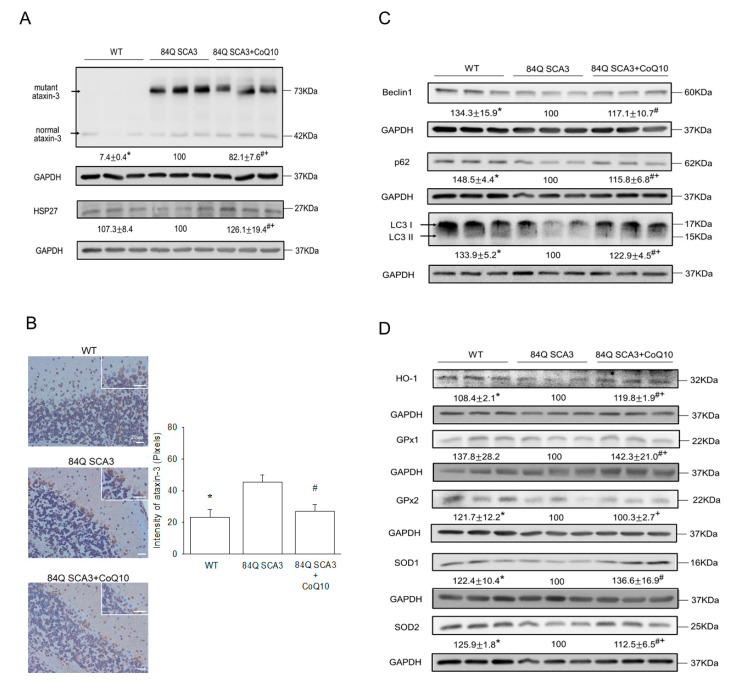
Effects of CoQ10 on the expression of mutant ataxin-3, Hsp27, autophagy and antioxidant protein expression in the cerebellum of 84Q SCA3 mice. (**A**) At the age of 16 months, the ataxin-3 expression of cerebellar extracts and (**B**) cerebellar tissue were analyzed by Western blot and immunobiological staining, respectively. Ataxin-3 expression in 50 Purkinje cells of the cerebellar cortex was quantified per group by collecting the magnification images shown in the upper right corner of the figure. (**A**) The expression of Hsp27 and (**C**) autophagy marker proteins, including p62, Beclin-1 and LC3-II, and (**D**) antioxidant proteins, including HO-1, GPx1, GPx2, SOD1 and SOD2 in cerebellar extracts, were analyzed by Western blot. All protein expressions were calculated by normalizing the internal control (GAPDH) prior to a comparison between groups. Values from treated mice were normalized to those of the 84Q SCA3 group. Data are presented as the mean ± SD. * *p* < 0.05 indicates the WT group compared to the 84Q SCA3 group. # *p* < 0.05 indicates the 84Q SCA3 + CoQ10 group compared to the 84Q SCA3 group. + *p* < 0.05 indicates the 84Q SCA3 + CoQ10 group compared to the WT group. WT, wild-type controls; 84Q SCA3, chow diet of vehicle; 84Q SCA3 + CoQ10, CoQ10 supplementation.

**Figure 6 nutrients-14-03593-f006:**
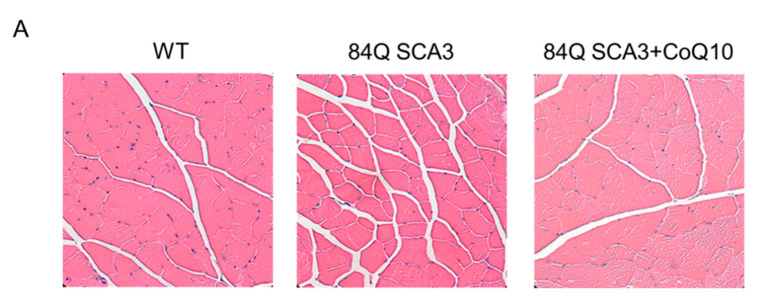
Effects of CoQ10 on muscle atrophy in the TA and GA muscles of 84Q SCA3 mice. At the age of 16 months, (**A**) graphs representing the H&E staining of TA muscle sections. (**B**) CSA of TA muscle fibers. (**C**) Pie charts representing the percentage with number of TA muscle fibers of CSA distribution: 0–1500, 1500–2000, 2000–2500, >2500 μm^2^. (**D**) The expression of MyHC and phosphorylated AKT in GA muscles normalized against an internal control (GAPDH) with the same amount of loading protein and the quantification shown as the down panel. In panels D, values from treated mice were normalized to those of the 84Q SCA3 group. Data are presented as the mean ± SD. * *p* < 0.05 indicates the WT group compared to the 84Q SCA3 group. # *p* < 0.05 indicates the 84Q SCA3 + CoQ10 group compared to the 84Q SCA3 group. + *p* < 0.05 indicates the 84Q SCA3 + CoQ10 group compared to the WT group. CSA, cross-sectional area; GA, gastrocnemius muscle; TA, tibialis muscle; WT, wild-type controls; 84Q SCA3, chow diet of vehicle; 84Q SCA3 + CoQ10, CoQ10 supplementation.

**Table 1 nutrients-14-03593-t001:** Effects of CoQ10 on the muscle mass/body weight ratio in 84Q SCA3 mice ^a^.

Treatment	WT	84Q SCA3	84Q SCA3 + CoQ10
Quadriceps muscle (mg/gw)	15.5 ± 1.3 *	12.5 ± 0.9	14.7 ± 0.5 ^#^
Tibialis muscle (mg/gw)	2.3 ± 0.6 *	1.4 ± 0.1	1.7 ± 0.3 ^#,^ ^+^
Gastrocnemius muscle (mg/gw)	11.0 ± 1.5 *	8.2 ± 1.5	9.9 ± 0.2
Soleus muscle (mg/gw)	1.4 ± 0.3 *	0.9 ± 0.1	1.1 ± 0.2 ^#^
Extensor digitorum longus muscle (mg/gw)	1.1 ± 0.1 *	0.8 ± 0.2	1.0 ± 0.2 ^#^

^a^ Data are presented as the mean ± SD. * *p* < 0.05, indicates the WT group compared to the 84Q SCA3 group. ^#^
*p* < 0.05, indicates the 84Q SCA3 + CoQ10 group compared to the 84Q SCA3 group. ^+^
*p* < 0.05 indicates the 84Q SCA3 + CoQ10 group compared to the WT group. WT, wild-type controls; 84Q SCA3, chow diet of vehicle; 84Q SCA3 + CoQ10, CoQ10 supplementation.

## Data Availability

The data presented in this study are available on request from the corresponding author.

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
