# Peer review of "Coenzyme Q10 Supplementation Increases Removal of the ATXN3 Polyglutamine Repeat, Reducing Cerebellar Degeneration and Improving Motor Dysfunction in Murine Spinocerebellar Ataxia Type 3"

_nutrients, 2022, doi:10.3390/nu14173593_

Round 1
Reviewer 1 Report
Thank you for the detailed paper. This is a CoQ10 study about SCA 3 in a mouse model. The introduction reads fine, although the authors are describing a mouse model, would suggest citing references to other human studies on ataxia disorders where there is evidence for CoQ10 e.g. Friedreich's ataxia. The methodology is well described. What was the reason age 16 months was chosen as the time to check CoQ10 levels. It is best for the authors not to comment on unpublished data. Overall interesting, however limitations on the subject. Thanks
Author Response
Point 1: Thank you for the detailed paper. This is a CoQ10 study about SCA 3 in a mouse model. The introduction reads fine, although the authors are describing a mouse model, would suggest citing references to other human studies on ataxia disorders where there is evidence for CoQ10 e.g. Friedreich's ataxia. The methodology is well described. What was the reason age 16 months was chosen as the time to check CoQ10 levels. It is best for the authors not to comment on unpublished data. Overall interesting, however limitations on the subject. Thanks
Response 1: Thanks for your comments to point out whether CoQ10 was applied on the human studies of ataxia disorders e.g. Friedreich's ataxia. We have added the related description in part of Discussion (page 14, line 436-439) with mentioned references to strengthen the clinical relevance with the present study. Briefly, high-dose CoQ10 improved the clinical score of Friedreich's ataxia patients until 47 months treatment. Otherwise, we found that the concentration of plasma CoQ10 was just detected at 7 months post-treatment (16 month old mice) (121.4±21.3 nM) but not in the beginning and short treatment. That is why we chosen the end-point at the 16 months old mice. On the other hand, we fully agree your opinion to avoid the comment of unpublished data. So, the benefit of exercise to amplify outcomes of CoQ10 treatment was added in the part of “Supplement”.
Reviewer 2 Report
Wu and Chang et al. explored the potential of using CoQ10 to treat SCA3 and investigated the mechanism. Here are my comments.
1. Were the 84Q SCA3 mice showing any phenotypes at the beginning of the treatment (9mon of age)? When is the starting time of cerebella degeneration and behavioral defects of this line? Were the transgenic mice smaller than the WT mice? If they were smaller, were there any body-size changes with CoQ10 treatment?
2. It’s not clear if the CoQ10 treated mice were more active than the untreated WT in the open-field behavioral test. In addition, the sex of the behavioral mice was not specified.
3. Cerebellar sections vary a lot depending on the distance from the vermis. No information was provided about how were the sections selected. Please provide a representative full-section image of each treatment group in Figure 2.
4. Figure 4B and 4D were confusing in that two areas were sampled, but only one bar graph was shown. Not sure if the measurements were combined or not. Also, Figure D legend claimed that ‘In all graphs, values from treated mice were normalized to those of the 84Q SCA3 group’. But the units of the bar graphs were µm, which indicates direct measurements. Please clarify.
5. The cerebellar granule cells are the most numerous neurons of the cerebellum. Using total protein extracts from homogenized cerebella in western blot, I don’t think any conclusion can be made about Purkinje cells, with the data shown in Figure 3A.
Author Response
Point 1: Were the 84Q SCA3 mice showing any phenotypes at the beginning of the treatment (9mon of age)? When is the starting time of cerebella degeneration and behavioral defects of this line? Were the transgenic mice smaller than the WT mice? If they were smaller, were there any body-size changes with CoQ10 treatment?
Response 1: The detailed phenotype of 84Q mice is showed on the website of the Jason laboratory (https://www.jax.org/strain/012705), the company we purposed.
In brief,
1) The mice have a mild and slowly progressive cerebellar deficit, manifesting as early as 4 weeks of age. As the disease progresses, pelvic elevation becomes markedly flattened, accompanied by hypotonia, and motor and sensory loss. Neuronal intranuclear inclusion (NII) formation and cell loss is prominent in the pontine and dentate nuclei, with variable cell loss in other regions of the cerebellum from 4 weeks of age. Interestingly, peripheral nerve demyelination and axonal loss is detected in symptomatic mice from 26 weeks of age.
2) The neurological phenotype is characterized by prominent gait abnormalities (~4 weeks), mild tremor, moderate hypoactivity, forelimb/hindlimb clasping (~24 weeks), inability to correct during negative geotaxis (~20 weeks), marked neuronal degeneration and mild gliosis of the dentate and pontine nerve nuclei, atrophy of the cerebellar Purkinje and molecular cell layers, and increased neuronal intranuclear inclusions (NIIs) that are ubiquitinated.
3) Mice have slower weight gain with a 24% weight reduction by 48 weeks of age compared to controls mice weigh 12% less than controls at weaning and 22% less at 20 weeks of age. Mice lose weight after 30 weeks of age. So, disease mice have the lower body weights but the difference is not significant from wild-type control mice at the start of treatment (9 mon=36 weeks). That could be the reason that CoQ10 treatment did not significantly affect the body weight shown as bellow.
|
Age / Body weight (gw) |
WT |
84Q SCA3 |
84Q SCA3+CoQ10 |
|
9 Months |
26.3±4.2 |
23.3±3.0 |
23.5±3.1 |
|
16 Months |
28.4±4.1 |
25.0±2.8 |
24.7±3.1 |
WT, wild-type controls; 84Q SCA3, chow diet of vehicle; 84Q SCA3+CoQ10, CoQ10 supplementation.
Point 2: It’s not clear if the CoQ10 treated mice were more active than the untreated WT in the open-field behavioral test. In addition, the sex of the behavioral mice was not specified.
Response 2: According to the result of Figure 1 of open-field behavior assay, mice local mobility between WT and treatment groups was similar with non-significant difference. The experiment data is calculated from a mix of male and females and related description has been mentioned in the revision (page 5, line 221-222).
Point 3: Cerebellar sections vary a lot depending on the distance from the vermis. No information was provided about how were the sections selected. Please provide a representative full-section image of each treatment group in Figure 2.
Response 3: The Figure 3 of sagittal section of the cerebellum near the junction of the vermis has been reorganized and consistently corrected to show the full-section image of each treatment group and further indicate the sampling area of III and IX lobules for analysis. The figure legend has been also modified.
Point 4: Figure 4B and 4D were confusing in that two are as were sampled, but only one bar graph was shown. Not sure if the measurements were combined or not. Also, Figure D legend claimed that ‘In all graphs, values from treated mice were normalized to those of the 84Q SCA3 group’. But the units of the bar graphs were µm, which indicates direct measurements. Please clarify.
Response 4: Sorry for the incorrect description to make you confusion. Only Figure 6D, values from treated mice were normalized to those of the 84Q SCA3 group. We have corrected the figure legend description (page 12, line376). Thank you for your careful correction.
Point 5: The cerebellar granule cells are the most numerous neurons of the cerebellum. Using total protein extracts from homogenized cerebella in western blot, I don’t think any conclusion can be made about Purkinje cells, with the data shown in Figure 3A.
Response 5: That is true so that related description in title of Figure 4A (page 9, line 299) and section 3.4 (page 8, line 286) was modified by replacing Purkinje cells with cerebellum in the revision.
Reviewer 3 Report
The Authors of this manuscript investigated the effects of a dietary supplementation with coenzyme Q10 in a transgenic mouse model of spinocerebellar ataxia type 3 (SCA3). Compared with untreated controls, they observed improved locomotion, normal stratificaiton of cerebellar layers, increased survival of Purkinje cell, as judged from reduced abundance of TUNEL-positive nuclei and activation of p53 and lower levels of cleaved caspase 3 and poly-ADP-ribose polymerase. They also observed increased expression of heat shock protein 27 and autophagic markers p62, Beclin-1 and LC3II, which they interpreted as a reflection of facilitated autophagy-mediated clearance of mutant ataxin-3 protein and restored expression of antioxidant enzymes. The Authors also observed significant increases in muscle fiber area and upregulated muscle protein synthesis pathways. They propose that on these grounds the usage of Q10 in SCA3 therapy might be considered.
Methodology
This paper has serious limitations in methodology.
1. The first, very important limitation is that the Authors did not measure the accumulation of ataxin-3 in Purkinje cells of untreated and treated 84Q SCA3 mice and of wild-type control animals, e.g., by employing immunohistochemistry with fluorescent antibodies. This was mandatory for the purpose of evaluating the efficacy of the treatment. In the absence of this kind of evidence, all conclusions drawn from the Authors are not sufficiently grounded. This is surprising, as the same research group has produced scientific reports in which they were able to perform this analysis (see, as an instance: Liu, SW., Chang, JC., Chuang, SF. et al. Far-infrared Radiation Improves Motor Dysfunction and Neuropathology in Spinocerebellar Ataxia Type 3 Mice. Cerebellum 18, 22–32 (2019). https://doi.org/10.1007/s12311-018-0936-3). The Authors should also have mentioned the pathological relevance and pathogenetic significance of the anomalous deposition of ataxin-3 in the 84Q SCA3 animal model of disease and in spinocerebellar ataxia in general.
2. Another important methodological limitation of this study is that there was not a control group of wild-type mice on a diet with Q10 supplementation, in addition to the control group of wild-type animals on a standard diet. This is not irrelevant and it is hard to understand why did the Authors choose to have three experimental groups (fourteen wild-type controls on chow diet, seven 84Q SCA3 mice on chow diet, and seven 84Q SCA3 mice with CoQ10 supplementation), instead of having four symmetrical groups of seven animals each (seven controls on chow diet, seven controls with CoQ10 supplementation, seven 84Q SCA3 mice on chow diet, seven 84Q SCA3 mice with CoQ10).
3. In all Figures, the Authors report that “values from treated mice were normalized to those of the 84Q SCA3 group”. In order to judge whether such normalization was acceptable, the Authors ought to clarify how was it done in a dedicated paragraph of Section 2.10 of the Methods, which should be renamed accordingly “Data normalization and statistical analysis”
4. As a general rule, the Authors should have indicated the statistical significance of all parameters measured and differences observed in 84Q SCA3 mice treated with CoQ10 not only in comparison with 84Q SCA3 mice, but also in comparison with wild-type control animals.
Abstract
Lines 37-38: “As a consequence, expression of antioxidant enzymes was restored”: please amend as follows: “The expression of antioxidant enzymes heme oxygenase 1 (HO-1), glutathione peroxidase 1 (GPx1), superoxide dismutase 1 (SOD1) and 2 (SOD2), but not of glutathione peroxidase 2 (GPx2), were restored in 84Q SCA3 animals treated with CoQ10 to levels even higher than those measured in wild-type animals”.
Results
Section 3.1. Effects of CoQ10 on plasma concentration, rotarod performance and open field behavior in 84Q SCA3 mice
This section should be split into two separate sections:
3.1. Plasma CoQ10 concentrations after a 7-month treatment
3.2. Rotarod performance and open field behavior of wild-type and 84Q SCA3 mice with and without CoQ10 supplementation
Figure 1 should and its legend should be split accordingly into two separate figures.
All subsequent sections and figures ought to be renumbered accordingly.
Figure 4.
- Please add the following explanation at the beginning of the legend: “Western blot analysis of total protein extracts of cerebellar homogenates”.
- Moreover, as the levels of GAPDH were quite unequal from one lane to another, the Authors should state explicitly whether they did the prescribed normalization of the other protein bands in the blot using GAPDH as a term of comparison.
Section 3.5. Effects of CoQ10 on skeletal muscle atrophy of 84Q SCA3 mice
- Lines 309-310: Please rephrase the sentence “In mice supplemented with CoQ10, these were significantly increased (Table 1), … although not to the levels observed in wild-type control animals” (THE AUTHORS PLEASE NOTICE: please indicate the statistical significance of CoQ10-treated versus wild-type controls).
- Lines 310-314: please rephrase as follows: “There was a significant decrease of TA fiber area (CSA) and size in 84Q SCA3 mice compared to WT mice, with 100% of TA muscle fibers having a CSA < 1500 μm2 (Figure 5A-B). CoQ10 treatment was associated with increased sizes of regenerated TA muscle fibers, whose percent distribution is shown in Figure 5C (THE AUTHORS PLEASE NOTICE: please indicate the statistical significance for each slice of the pie charts). The expression of MyHC in gastrocnemius (GA) muscle was also significantly increased in CoQ10-treated animals in comparison with diseased animals (Figure 5D; p<0.05), although not to the level of wild-type control animals (THE AUTHORS PLEASE NOTICE: please indicate the statistical significance of CoQ10-treated versus wild-type controls).”
Figure 5.
Considering the poor quality of triplicates in Figure 5D, particularly with regard to the p-AKT bands, the Authors are requested to add a bar diagram clearly showing the amplitude of the SDs, which presumably was the reason why the difference between treated animals and diseased animals was not statistically significant. Please also indicate the statistical significance of CoQ10-treated versus wild-type controls.
Author Response
The Authors of this manuscript investigated the effects of a dietary supplementation with coenzyme Q10 in a transgenic mouse model of spinocerebellar ataxia type 3 (SCA3). Compared with untreated controls, they observed improved locomotion, normal stratificaiton of cerebellar layers, increased survival of Purkinje cell, as judged from reduced abundance of TUNEL-positive nuclei and activation of p53 and lower levels of cleaved caspase 3 and poly-ADP-ribose polymerase. They also observed increased expression of heat shock protein 27 and autophagic markers p62, Beclin-1 and LC3II, which they interpreted as a reflection of facilitated autophagy-mediated clearance of mutant ataxin-3 protein and restored expression of antioxidant enzymes. The Authors also observed significant increases in muscle fiber area and upregulated muscle protein synthesis pathways. They propose that on these grounds the usage of Q10 in SCA3 therapy might be considered.
Methodology
This paper has serious limitations in methodology.
Point 1: The first, very important limitation is that the Authors did not measure the accumulation of ataxin-3 in Purkinje cells of untreated and treated 84Q SCA3 mice and of wild-type control animals, e.g., by employing immunohistochemistry with fluorescent antibodies. This was mandatory for the purpose of evaluating the efficacy of the treatment. In the absence of this kind of evidence, all conclusions drawn from the Authors are not sufficiently grounded. This is surprising, as the same research group has produced scientific reports in which they were able to perform this analysis (see, as an instance: Liu, SW., Chang, JC., Chuang, SF. et al. Far-infrared Radiation Improves Motor Dysfunction and Neuropathology in Spinocerebellar Ataxia Type 3 Mice. Cerebellum 18, 22–32 (2019). https://doi.org/10.1007/s12311-018-0936-3). The Authors should also have mentioned the pathological relevance and pathogenetic significance of the anomalous deposition of ataxin-3 in the 84Q SCA3 animal model of disease and in spinocerebellar ataxia in general.
Response 1: Express of ataxin-3 in Purkinje cells was visualized by IHC staining and quantified by randomly counting 50 per group (N=3). Please refer to revised Figure 5B. Thanks.
Point 2: Another important methodological limitation of this study is that there was not a control group of wild-type mice on a diet with Q10 supplementation, in addition to the control group of wild-type animals on a standard diet. This is not irrelevant and it is hard to understand why did the Authors choose to have three experimental groups (fourteen wild-type controls on chow diet, seven 84Q SCA3 mice on chow diet, and seven 84Q SCA3 mice with CoQ10 supplementation), instead of having four symmetrical groups of seven animals each (seven controls on chow diet, seven controls with CoQ10 supplementation, seven 84Q SCA3 mice on chow diet, seven 84Q SCA3 mice with CoQ10).
Response 2: Considering that normal individual doesn’t need to take high dose CoQ10 and animal welfare, clarification of CoQ10’s effect is indeed partially restricted in our study. We hope you understand that. On the other hand, sorry for typing error number of mice in WT group and it has been corrected to “n=7” (page 3, line 132) in the methods of the revision. Thanks.
Point 3: In all Figures, the Authors report that “values from treated mice were normalized to those of the 84Q SCA3 group”. In order to judge whether such normalization was acceptable, the Authors ought to clarify how was it done in a dedicated paragraph of Section 2.10 of the Methods, which should be renamed accordingly “Data normalization and statistical analysis”
Response 3: Indeed, sorry for unclear statement of statics to make you confuse. Considering to your suggestion, we corrected and added the related description of statistic index in each figures legend of revision. Thanks for your correction.
Point 4: As a general rule, the Authors should have indicated the statistical significance of all parameters measured and differences observed in 84Q SCA3 mice treated with CoQ10 not only in comparison with 84Q SCA3 mice, but also in comparison with wild-type control animals.
Response 4: Thanks for your comments and considering to easy reading for readers, we added the symbol “+” labeling with related description of statistic index to show the statistic difference between CoQ10-treated 84Q SCA3 mice and with wild-type control animals in each figure and figure legend of the revision.
Abstract
Point 1: Lines 37-38: “As a consequence, expression of antioxidant enzymes was restored”: please amend as follows: “The expression of antioxidant enzymes heme oxygenase 1 (HO-1), glutathione peroxidase 1 (GPx1), superoxide dismutase 1 (SOD1) and 2 (SOD2), but not of glutathione peroxidase 2 (GPx2), were restored in 84Q SCA3 animals treated with CoQ10 to levels even higher than those measured in wild-type animals”.
Response 1: The sentence in Abstract has been modified as your suggestion in the revision (page 1, line 44-48). Thanks.
Results
Point 2: Section 3.1. Effects of CoQ10 on plasma concentration, rotarod performance and open field behavior in 84Q SCA3 mice
This section should be split into two separate sections:
3.1. Plasma CoQ10 concentrations after a 7-month treatment
3.2. Rotarod performance and open field behavior of wild-type and 84Q SCA3 mice with and without CoQ10 supplementation
Figure 1 should and its legend should be split accordingly into two separate figures.
All subsequent sections and figures ought to be renumbered accordingly.
Response 2: Presented separation of Section 3.1. and Figure 1 have been executed according to your suggestion and the subsequent sections and figures have been renumbered accordingly (page 5, line 226- page 7, line 265). Thanks.
Point 3: Figure 4.
1.- Please add the following explanation at the beginning of the legend: “Western blot analysis of total protein extracts of cerebellar homogenates”.
Response 3-1: Thanks for your suggestion and the sentence has been added “Western blot analysis of total protein extracts of cerebellar homogenates” in the legend of revision (page 9, line 300).
2.- Moreover, as the levels of GAPDH were quite unequal from one lane to another, the Authors should state explicitly whether they did the prescribed normalization of the other protein bands in the blot using GAPDH as a term of comparison.
Response 3-2: Thanks for your careful correction. All proteins in the western blot have been normalized by its internal controls (GAPDH) prior to comparison of different protein levels within groups in the blot. The related description has been added in the legend of figures of revised edition (Page 11, line 341-342).
3. Section 3.5. Effects of CoQ10 on skeletal muscle atrophy of 84Q SCA3 mice
- Lines 309-310: Please rephrase the sentence “In mice supplemented with CoQ10, these were significantly increased (Table 1), … although not to the levels observed in wild-type control animals” (THE AUTHORS PLEASE NOTICE: please indicate the statistical significance of CoQ10-treated versus wild-type controls).
Response 3-3: The statistical significance of CoQ10-treated versus wild-type controls was labeled by the symbol “+” labeling in revised Table 1 with the related description of statistic index in the legend of Table 1 (page 11, line 350-351.
4. Section 3.5.
- Lines 310-314: please rephrase as follows: “There was a significant decrease of TA fiber area (CSA) and size in 84Q SCA3 mice compared to WT mice, with 100% of TA muscle fibers having a CSA < 1500 μm2 (Figure 5A-B). CoQ10 treatment was associated with increased sizes of regenerated TA muscle fibers, whose percent distribution is shown in Figure 5C (THE AUTHORS PLEASE NOTICE: please indicate the statistical significance for each slice of the pie charts). The expression of MyHC in gastrocnemius (GA) muscle was also significantly increased in CoQ10-treated animals in comparison with diseased animals (Figure 5D; p<0.05), although not to the level of wild-type control animals (THE AUTHORS PLEASE NOTICE: please indicate the statistical significance of CoQ10-treated versus wild-type controls).”
Response 3-4: Thanks for your detailed correction. The standard deviation of TA muscle fiber sizes and statistical significance between each group have been added in the pie charts of revised Figure 6C. The sentence also has been modified as “There was a significant decrease of TA fiber area (CSA) and size in 84Q SCA3 mice compared to WT mice, with 100% of TA muscle fibers having a CSA < 1500 μm2 (Figure 6A-B). CoQ10 treatment was associated with increased sizes of regenerated TA muscle fibers, whose percent distribution is shown in Figure 6C. The expression of MyHC in gastrocnemius (GA) muscle was also significantly increased in CoQ10-treated animals in comparison with diseased animals (Figure 6D; p<0.05), although not to the level of wild-type control animals " in the revision (page 11, line 358-364). The symbol “+” labeling was used to show a statistic difference between treated disease mice and wild-type control mice in the revised Figure 6 B and D and related description of statistic index was also added in each revised legend.
Point 4: Figure 5.
Considering the poor quality of triplicates in Figure 5D, particularly with regard to the p-AKT bands, the Authors are requested to add a bar diagram clearly showing the amplitude of the SDs, which presumably was the reason why the difference between treated animals and diseased animals was not statistically significant. Please also indicate the statistical significance of CoQ10-treated versus wild-type controls.
Response 4: Sorry for the poor quality of blot images. We checked and recalculated the raw data and found the significant difference between treated animals and diseased animals. We have corrected the result in revised Figure 6D and also compared the difference of treated disease mice and with wild-type control (non -difference). Thanks.
Round 2
Reviewer 2 Report
Thank you for working on my comments.
Author Response
Point 1: Thank you for working on my comments.
Response 1: Thanks, this is my business.
Reviewer 3 Report
The Authors made changes to the manuscript, in acceptance of my comments, which substantially improved the quality of the paper. The Authors’ responses were not entirely satisfactory in two cases, which are detailed below:
- in regard to the Authors’ response 2 to my comments on the methodology of the study, I’m willing to be understanding concerning the restrictions of the Q10 supplementation in normal animals. However, the numbers in lines 131-133 still don’t add up: “28 experimental mice were randomly divided into three groups: wild-type (WT) mice (n=7); 84Q SCA3 mice with chow diet (n=7) and 84Q SCA3+CoQ10 (n=7).” Now, 3 times 7 is 21. So, where is the mistake? Were the experimental mice 21 rather than 28, or were the WT mice actually 14 (if they were, so they be, no problem with this);
- in regard to the Authors’ response to my comment on Figure 5, I can’t help being surprised that a difference between the p-AKT expression in the 84Q SCA3+CoQ group compared to the 84Q SCA3 group that initially was not statistically significant, then turned out to be significant upon recalculation of the raw data. Please, do not get me wrong: I am not implying any malice. Just check it over a second time. By making my comment, I just meant what I meant, i.e., that the variation between replicates in the blots was wide, which was probably the reason for the lack of statistical significance in the difference. Thus, what the Authors had seen was to be taken as a trend rather than as a sheer difference, which demanded some caution. For the sake of clarity, this is a kind of data which demands a graphical representation as transparent and pictorial as possible. Hence my request for the addition of a bar diagram with error bars, to be added to better show the differences between groups in the blots shown in panel D of Figure 6, which I maintain.
Author Response
The Authors made changes to the manuscript, in acceptance of my comments, which substantially improved the quality of the paper. The Authors’ responses were not entirely satisfactory in two cases, which are detailed below:
Point 1: in regard to the Authors’ response 2 to my comments on the methodology of the study, I’m willing to be understanding concerning the restrictions of the Q10 supplementation in normal animals. However, the numbers in lines 131-133 still don’t add up: “28 experimental mice were randomly divided into three groups: wild-type (WT) mice (n=7); 84Q SCA3 mice with chow diet (n=7) and 84Q SCA3+CoQ10 (n=7).” Now, 3 times 7 is 21. So, where is the mistake? Were the experimental mice 21 rather than 28, or were the WT mice actually 14 (if they were, so they be, no problem with this)
Response 1: Thanks for your detailed correction. You are right. Sorry for typing error number of mice in total experimental mice and it has been corrected to “21” (page 3, line 131) in the methods of the revision. Thanks.
Point 2:in regard to the Authors’ response to my comment on Figure 5, I can’t help being surprised that a difference between the p-AKT expression in the 84Q SCA3+CoQ group compared to the 84Q SCA3 group that initially was not statistically significant, then turned out to be significant upon recalculation of the raw data. Please, do not get me wrong: I am not implying any malice. Just check it over a second time. By making my comment, I just meant what I meant, i.e., that the variation between replicates in the blots was wide, which was probably the reason for the lack of statistical significance in the difference. Thus, what the Authors had seen was to be taken as a trend rather than as a sheer difference, which demanded some caution. For the sake of clarity, this is a kind of data which demands a graphical representation as transparent and pictorial as possible. Hence my request for the addition of a bar diagram with error bars, to be added to better show the differences between groups in the blots shown in panel D of Figure 6, which I maintain.
Response 2:The present result of Fig 6D has been confirmed and the raw data of the p-AKT, attached as bellow, indeed showed the significance between 84Q SCA3+CoQ group and disease group. Thanks.
"Please see the attachment."
